# Morphological Analysis of a New Species of *Micropterus* (Teleostei: Centrarchidae) from Lake Erie, PA, USA

**DOI:** 10.3390/biology13090660

**Published:** 2024-08-26

**Authors:** Andrew T. Ross, Jay R. Stauffer

**Affiliations:** 1Ecosystem Science and Management, The Pennsylvania State University, University Park, PA 16802, USA; aur309@psu.edu; 2South African Institute for Aquatic Biodiversity, Makhanda 6140, South Africa

**Keywords:** black basses, species concepts, Lake Erie

## Abstract

**Simple Summary:**

A new species of *Micropterus* (Black Basses) is described from Lake Erie. This species was historically thought to be the Largemouth Bass, *Micropterus nigricans*. It is distinguished from previously described *Micropterus* spp. by morphological characters.

**Abstract:**

This previously undescribed species of *Micropterus* is described from collections in Lake Erie. The species was first recognized by the tri-colored tail of juveniles. This tri-colored tail, the presence of bold parallel lines ventral to the lateral band, and scales in the inter-radial membranes of the dorsal fin distinguish it from *Micropterus nigricans*. The large mouth that extends past the eye distinguishes it from *Micropterus punctulatus*.

## 1. Introduction

Unique phenotypes are used to delimit lineages and thus recognize evolutionary species [1], and morphological analyses have been used to diagnose undescribed fishes [2,3]. The evolutionary synthesis certainly attempted to form and unify the theory of evolution [4] and resulted in a re-evaluation of the meaning of the term species [5]. To date, there is still no good consensus of what constitutes a species [5].

The genus *Micropterus*, also known as Black Basses, is widely distributed throughout North America, with some species endemic to restricted areas. The genus is delimited by a combination of three anal-fin spines and greater than 55 lateral-line scales [6]. *Micropterus* is differentiated from other centrarchids by these traits; however, they are grouped with the sunfishes (*Lepomis* spp.) in the subfamily Lepominae [7], which also have three anal-fin spines but fewer than 55 lateral-line scales. *Micropterus* have 46 chromosomes, whereas *Lepomis* spp. have 48 [8]. Within *Micropterus*, *Micropterus nigricans*, Largemouth Bass, is ancestral to the other species in the genus. Closely related to the Largemouth Bass is the Spotted Bass, *Micropterus punctulatus*.

A phylogenetic review [9] found that a nomenclatural revision within *Micropterus* was necessary. The specific epithet of the Largemouth Bass was changed to reflect that *Micropterus salmoides* is the earliest name used for the Florida Bass; thus, the correct scientific name of the Florida Bass was changed from *Micropterus floridanus* to *M. salmoides*. This change resulted in *M. salmoides* becoming a junior synonym for the Largemouth Bass that occurs in the northern part of the United States. The earliest name used for the northern species was *M. nigricans*; thus, the scientific name for the Largemouth Bass became *Micropterus nigricans* [9].

*Micropterus nigricans* is native to the Mississippi River basin and is found from Florida to Northeastern Mexico as well as north in the Great Lakes drainages of Southern Canada [6]. In the Atlantic Slope, the range of the Florida Bass is limited to southern Florida, and southern and central South Carolina [10]. In Pennsylvania, the Largemouth Bass is indigenous to the Ohio River, although it has been introduced throughout the commonwealth and now occurs in the Potomac, Delaware, and Susquehanna drainages [6]. The Largemouth Bass is characterized and differentiated from other centrarchids by a large, slightly sloping mouth, and a horizontal mid-lateral stripe. The corner of the upper jaw extends past the posterior edge of the eye. Teeth occur on the upper and lower jaws, but no teeth are found on the tongue. Largemouth Bass possess 10–11 dorsal-fin spines, 12–13 dorsal-fin rays, 10–11 anal-fin rays, and, typically, 60–68 lateral-line scales [11]. Scales located along the dorsum or base of the dorsal fin do not encroach onto the dorsal fin. Pyloric caeca are typically branched in Largemouth Bass. 

The Largemouth Bass inhabits both lentic and lotic habitats but appears to prefer clear, slow water and uses solar cues for orientation [12]. Abundance is positively associated with submerged vegetation and debris [13,14]. In areas with reduced vegetation, Largemouth Bass with increased growth are seen more frequently. This is due to their young becoming carnivorous at smaller sizes [15]. Largemouth Bass are selective feeders [16] and use an ambush strategy to prey mainly on minnow species, but actively hunt for sunfish species [17]. 

*Micropterus punctulatus* is native to the Mississippi River basin ranging from southern Ohio, south to the Gulf of Mexico [6]. It is also native to the Gulf Slope drainages including the Chattahoochee River and the Guadalupe River [18]. Characterized by a tooth patch on its tongue, the Spotted Bass is differentiated from other similar *Micropterus* spp. lacking such a patch. The posterior edge of the maxilla reaches to and occasionally beyond the posterior edge of the eye. Young-of-the-year Spotted Bass exhibit a tri-colored tail and a more interrupted lateral band in juveniles. Lateral-line scale count ranges between 60 and 68 with some individuals possessing as few as 55 and as many as 77 [19]. Additionally, Spotted Bass possess 9–10 dorsal-fin spines, 11–13 dorsal-fin rays, 10 anal-fin rays, and 15–16 pectoral-fin rays [6]. Pyloric caeca are typically unbranched. Lateral scales encroach on the dorsal fin.

Spotted Bass inhabit lotic environments such as large streams and rivers [20]. They are strongly associated with large woody debris, natural root wads, and undercut banks [21]. Schooling behavior is also more commonly associated with Spotted Bass as opposed to Largemouth Bass. Spawning occurs around springtime when water temperatures reach 16–20 °C. Males are responsible for nest construction and guarding. Nests are made over gravel or rocks and are located typically in large, submerged cover such as logs or brush. Once the eggs are laid by the female, the male will guard the eggs and the fry. Spotted Bass search for food mostly in benthic areas with vegetation [22]. They feed primarily on crayfish, followed by small fishes and larval insects [23]. Juveniles and young feed on small crustations and midge larvae. 

In 2012, the Choctaw Bass, *Micropterus haiaka*, was discovered in Holmes Creek, Florida (Florida FWC). This species is deemed distinct from the Spotted Bass and Alabama Bass, *Micropterus henshalli*, via a genetic study completed by a Fish and Wildlife Research Institute involving other Black Bass [24]. The Choctaw Bass did not genetically match any currently described *Micropterus* species. As of 2012, Spotted Bass and Alabama Bass were not found in locations in which Choctaw Bass were collected (myfwc.com). 

Our purpose is to describe a new species of *Micropterus* from Misery Bay and Thompson Bay in Presque Isle State Park, Erie Pennsylvania (Figure 1). The new species is compared to other populations of *Micropterus* spp. 

## 2. Materials and Methods

A total of 63 fish from 13 collections from the Penn State University Fish Museum was examined (IACUC 436460-1). Twenty-seven of these individuals were the undescribed species from Thompson Bay and Misery Bay. Nine individuals of *Micropterus* spp. were from Long Point, Canada, 10 *M. punctulatus* from West Virginia, 10 *M. nigricans* from the Allegheny River, PA, and seven *M. nigricans* from the introduced population in the Susquehanna River PA. All fish collected were fixed in 10% formalin and preserved in 70% ethanol. For each fish, 24 measurements and 15 counts [25] were taken. When possible, 10 individuals were measured per collection from each locality. Five collections were from Presque Isle Bay (Misery Bay and Thompson Bay) in Lake Erie. One collection was taken from a locality near Long Point Canada, Lake Erie. Five collections from the Ohio River drainage in West Virginia of *M. punctulatus* (PSU 10019, PSU 10039, PSU 7077, PSU 1753, PSU 1751) were analyzed. Two collections of *M. nigricans* were compared. Of these, one was an introduced population from Marsh Creek (Susquehanna River Drainage) Pennsylvania (PSU 6082) and a native population from the Ohio River Drainage, Green County, PA (PSU 758).

All counts and measurements were made on the left side of the fish except for the gill-raker counts, which were collected from the right side. Measurements included: standard length (SL) tip of the snout to the posterior end of the hypural plate; head length from the tip of the snout to the end of the operculum; snout length (SNL) from the tip of the snout to the anterior orbital; post-orbital head length (POHL) from the anterior orbital to the anterior insertion of first dorsal fin; horizontal eye diameter (HED) and vertical eye diameter (VED); premaxillary pedicel length (PRE); lower jaw length (LJL); head depth (HD) from the junction of branchiostegal rays to top of head; interior orbital width (IOWI) distance between the two eyes; from tip of the snout to anterior insertion of the first dorsal fin (SNDOR); snout to anterior insertion of pelvic fin (SNPEL); length of the first dorsal fin (1DFBL); length of the second dorsal fin (2DFBL); anterior insertion of the first dorsal fin to the anterior insertion of anal fin (ADAA); anterior insertion of the second dorsal fin to the posterior insertion of the anal fin (ADPA); posterior insertion of second dorsal fin to anterior insertion of the anal fin (PDAA); posterior insertion of the second dorsal fin to the posterior insertion of anal fin (PDPA); posterior insertion of the second dorsal fin to the ventral point of hypural plate (PDVC); posterior insertion of anal fin to the dorsal portion of hypural plate (PADC); anterior insertion of second dorsal fin to the anterior insertion pelvic fin (ADP2); posterior insertion of first dorsal fin anterior insertion of the pelvic fin (PDP2); mid-lateral point above posterior insertion of the anal fin to hypural plate (CPL); and narrowest point of the caudle peduncle (LCPD). Meristic data included: dorsal-fin spines (DSPINES); dorsal-fin rays (DRAYS); anal-fin spines (ASPINES); anal-fin rays (ARAYS); pectoral-fin rays (P1RAYS); pelvic-fin rays (P2RAYS); lateral-line scales (LLS); check-scale rows (CFS); operculum scale rows (OSR); preopercular scale rows (PREOSR); gill-rakers on epibranchial (GRUP); gill-rakers on outer row of ceratobranchial (GRLOW); number of brachial pores (BRPORES); and total rays (number of anal rays + dorsal rays, TR).

A holotype specimen was designated from a collection from Thompson Bay, 2022.2 Lake Erie and was CT scanned. Scanning was completed via The Center for Quantitative Imaging’s high-resolution industrial MicroCT scanner, a GE Pheonix V|Tome|X L. Using these CT scan images, the processed slices were then compared using Avizo 3D 2022.2 to analyze and isolate the 3D slices. The reconstructions from these scans are to be uploaded to morphosource.

Analyses of morphometrics and meristics were conducted using sheared principal component analysis (SPCA) and principal component analysis (PCA), respectively, as described by Humphries et al. [26] and Stauffer et al. [27]. Principal components analysis (PCA) was used to analyze meristic data with the correlation matrix factored. Body-shape differences were analyzed using SPCA with the covariance matrix factored. To illustrate differences in counts and measurements among species, the sheared second principal components (SHRD PC2) of the morphometric data were plotted against the first principal components of the meristic data. Minimum polygon clusters were drawn to encompass the points of a population on the principal component plots. The first sheared principal component of the morphometric data accounted for the variation in individual size. Similarly, the sheared second principal components explained the remaining variation in shape. An ANOVA was used to determine if the mean multivariate scores were significantly different (*p* < 0.05) along one axis independent of the other. Subsequently, a Duncan’s Multiple Range Test was used to determine which clusters were significantly different from each other [26]. Stauffer et al. [27] used a similar analysis to delimit 10 species of *Metriaclima* in Lake Malawi.

## 3. Results

*Micropterus parallelus*, n. sp. (Figure 2 and Figure 3)

*Micropterus salmoides* (Lacépéde) (Stauffer et al. 2016)

*Micropterus nigricans* (Cuvier) 

### 3.1. Holotype

PSU 7410, adult male, 190.9 mm SL, Thompson Bay, Presque Isle, Lake Erie, Erie, PA (42.166k73, −80.082437)

Paratypes., data as for holotype, Thompson Bay. Six specimens PSU 7420 10, 128.1–307.0 mm SL, Misery Bay, Presque Isle, Lake Erie, Erie, PA, 42.153703, −80.112641, 19 specimens 82.8–321.6 mm SL, PSU 7410, 21, PSU 7416.

### 3.2. Diagnosis

The moderately compressed elongate body with more than 55 lateral-line scales, a mouth that extends under the eye, and 9–10 dorsal-fin spines places *M. parallelus* in the genus *Micropterus*. The large mouth that extends past the eye distinguishes *M. parallelus* from *Micropterus treculii* (Vaillant and Boucourt), *Micropterus dolomieu* (Lacépéde), and *M. punctulatus* (Rafinesque). It is distinguished from *Micropterus coosae* (Hubbs and Bailey), *Micropterus cahabae* (Baker, Blanton Johnston), *Micropterus tallapoosae* (Baker, Blanton Johnston), *Micropterus chattahoochae* (Baker, Blanton Johnston), and *Micropterus warriorensis* (Baker, Blanton Johnston), which all lack a dark mid-lateral stripe [28]. *Micropterus cataractae* (Willams & Burgess) has 67–81 lateral-line scales (typically 72–77) [29] and *Micropterus henshalli* (Hubbs and Bailey) has 68–84 [30,31]], while *M. parallelus* has 51–68, with only one specimen having 68, and the remainder with 66 or fewer lateral-line scales. Additionally, both *M. henshalli* and *Micropterus notius* (Bailey and Hubbs) have weakly developed lines ventral to the lateral-line [29,30], while *M. parallelus* has bold parallel lines ventral to the lateral-line. *M. nigricans* and *M. salmoides* lack scales on the inter-radial membranes of the dorsal fin, which are present in *M. parallelus*. *Micropterus parallelus* has a tooth patch on the basihyal, which is absent in *Micropterus* spp. collected from Lake Erie near Long Point, Canada, *M. nigricans*, and *M. floridanus* [31].

### 3.3. Description

Meristic and morphometric data in Table 1. Elongate species with greatest body depth at base of fourth dorsal-fin spine. Dorsal body profile with gradual curve to caudal peduncle. Ventral body profile between pelvic fins and anal fin flat with upward curve to caudal fin. Lateral body with series of parallel lines below lateral line. Jaw to posterior to eye. Number of dorsal-fin spines 9–10, dorsal-fin rays 11–13, anal-fin spines 3, and anal-fin rays 10–12Lateral scales large (57–68).

Dorsally head dark green/brown; black opercular spot. Laterally, dorsal one-quarter dark green/brown; ventral one-quarter cream color with distinct black lateral bands. Ventral pale yellow/white. Anal fin gray proximally with dark gray band on distal one-third; caudal find light gray proximally, dark gray distally.

### 3.4. Etymology 

The specific epithet parallelus is an adjective and refers to the bold parallel lines located ventrally of the lateral line. The suggested common name is Presque Isle Bass.

### 3.5. Distribution

*Micropterus parallelus* is restricted to Presque Isle Bay, Lake Erie, Pennsylvania (Figure 1).

### 3.6. Remarks

Nine individuals of *Micropterus* spp. were collected in Lake Erie from Long Point, Canada (42.582877, −80.436170) (Table 2). 

The minimum polygon clusters formed when the first principal components of the meristic data (PC1) are plotted against the second sheared principal components (SHRD PC2) of the morphometric for all populations from Lake Erie are shown in Figure 4. 

The first principal component (size variable) of the morphometric data (SHRD PC2) explained 90% of the observed variance, with the sheared second principal component accounting for 29% of the remaining. Variables that had the highest loadings on SHRD PC2 were the distance between the posterior insertion of the anal fin and the dorsal insertion of the caudal fin (−0.42), the head depth (0.36), and the caudal peduncle length (−0.28). The first principal component of the meristic data explained 39% of the variance. Variables with the highest loadings on the first principal components of the meristic data were preopercular scale rows (0.60), cheek-scale rows (0.47), and opercular scale rows (0.42). Analysis of variance in conjunction with a Duncans Multiple Range test showed that all data from populations collected at Long Point, Canada, were significantly different (*p* < 0.05) along the PC1 (meristic data) axis from the two populations of *M. parallelus*, but that data from the two populations of *M. parallelus* were not significantly different (*p* < 0.05) from each other.

Morphological and meristic data for *M. punctulatus* are summarized in Table 3.

Because of the morphological similarities between *M. parallelus* and *M. punctulatus* (i.e., tri-colored tail of juveniles, scales that encroach onto the dorsal fin, tooth patch on the basihyal), we examined the minimum polygon clusters formed when the first principal components of the meristic data (PC1) were plotted against the second sheared principal components (SHRD PC2) of the morphometric data (Figure 5).

There was no overlap in the clusters formed by data from *M. parallelus* and *M. punctulatus*. The first principal component (size variable) of the morphometric data (SHRD PC2) explained 99% of the observed variance with sheared PC2 accounting for 9.9% of the remaining. Variables that had the highest loadings on SPCA2 were the size of the eye (VED= −0.56, HED = −0.45), distance between the posterior insertion of the anal fin and the dorsal insertion of the caudal fin (0.28), and caudal peduncle length (0.24). The first principal component of the meristic data explained 33% of the variance. Variables with the highest loadings on the first principal components of the meristic data were anal-fin rays (0.49), dorsal-fin rays (0.44), and opercular scale rows (0.44). 

*Micropterus parallelus* was historically identified as *M. nigricans* and, in fact, the large mouth that extends past the eye is a character state found in both species. The minimum polygon clusters formed when the first principal components of the meristic data (PC1) are plotted against the second sheared principal components (SHRD PC2) of the morphometric data for these two species are shown in Figure 6. 

Analysis of variance in conjunction with a Duncans Multiple Range test showed that all three populations were significantly different (*p* < 0.05) along the sheared PC2 axis. Data from *M. parallelus* were significantly different (*p* < 0.05) from both populations of *M. nigricans*, which were not significantly different from each other. The first principal component (size variable) of the morphometric data (SHRD PC2) explained 99% of the observed variance with SHRD PC2 accounting for 26% of the remaining variance. Variables that had the highest loadings on SPCA2 were posterior insertion of the anal fin to dorsal insertion of the caudal fin (0.37), lower jaw length (−0.35), and head depth (0.33). The first principal component of the meristic data explained 52% of the variance. Variables with the highest loadings on the first principal components of the meristic data were preopercular scale rows (0.55), opercular scale rows (0.46), and anal-fin rays 45). 

*Micropterus* spp. from Long Point, Lake Erie, Canada lacks a tooth patch while M. parallelus has a tooth patch on the basihyal; thus, we examined other morphological characteristics. The minimum polygon clusters formed when the first principal components of the meristic data (PC1) are plotted against the second sheared principal components (SHRD PC2) of the morphometric for all populations from Lake Erie are shown in Figure 7. The minimum polygon clusters of both populations of M. parallelus do not overlap with those from the population at Long Point, Canada.

The first principal component (size variable) of the morphometric data (SHRD PC2) explained 99% of the observed variance with sheared PC2 accounting for 13.1% of the remaining. Variables that had the highest loadings on SPCA2 were size of the eye (VED = −0.62, HED = −0.47), head depth (0.25), and snout length (−0.20). The first principal component of the meristic data explained 96% of the variance. Variables with the highest loadings on the first principal components of the meristic data were anal-fin rays (0.50), dorsal-fin rays (0.46), and opercular scale rows (0.46). 

## 4. Discussion

As noted by Stauffer et al. [32], the definition of a species has stimulated more discussions and arguments than perhaps any other topic in evolutionary biology [33]. There are a minimum of 22 different species concepts [34]. Nelson [35] combined segments of the Evolutionary Species Concept [36,37] and the Biological Species Concept [38] to form the Evolutionary Biological Species Concept, which are groups of reproductively isolated populations that are evolutionary lineages diagnosed by irreversible discontinuities. If we regard species as ontological individuals, sensu Ghiselin [39], we cannot define them, and hence, the boundaries between two species may be fuzzy [40,41]. Nevertheless, species are the currency of biodiversity [34]; thus, they must be diagnosed and recognized if we are to effectively conserve ecosystems. 

Historically, many fishes are described based on morphological characters [2,3,27,42]. The analysis of morphometric data has changed from the use of univariate morphometric analysis to the quantification of shape [26]. These techniques ordinate the morphometric factors so that they are independent of a main linear ordination [26,43]. Such techniques have improved the resolution of morphometric distinctions, but color patterns of similar species have been used to delimit species [44]. Herein, we used a combination of morphological characters (e.g., HL) and pigmentation (e.g., parallel horizontal bars below the lateral line) to diagnose *M. parallelus*.

For the most part, history-based concepts use character analysis to reveal groups of individuals that qualify as basal evolutionary units or species [45]. It is the existence of unique phenotypes, not observations of reproductive isolation, that is the primary criterion upon which taxonomists recognize species [46,47]. Therefore, if allopatric populations have evolved one or more derived characters, then the evolutionary modification of ancestral characters has been satisfied and the population should be given species status [48,49]. Such criteria were used when many species were described from Lake Malaŵi [2,3,25]

We have described *M. parallelus* from Presque Isle, Lake Erie. Based on morphological data, the new species is distinct from all other known *Micropterus*. The population from Lake Erie at Long Point, Canada, is problematic. It differs in that a tooth patch on the basihyal is absent, while it is present in *M. parallelus*. Furthermore, it is morphologically distinct from *M. nigricans* as shown in Figure 7. Hybridization between *M. parallelus* and the population at LongPoint within Lake Erie may be possible and has been reported between other species of *Micropterus* [50].

Morphologically, the introduced and native populations of *M. nigricans* from Pennsylvania are similar, but there are some noted differences including that the head depth expressed as percent head length does not overlap (7.9–8.3% HL vs. 8.5–9.6% HL, Table 3). This difference may be due to the small sample size or introgression with the Florida Bass in the introduced population

*Micropterus punctulatus* has never been reported from the Great Lakes [6], although certain morphological features (e.g., tooth patch on basihyal, encroachment of scales on the dorsal fin, tri-colored tail in juveniles) are found in *M. parallellus*, although the two species can be easily distinguished (Figure 5). These similarities may have occurred via parallel evolution. The fact, however, that juveniles of *Micropterus dolomieu*, one of the earliest diverged species in the genus [51], also has a tri-colored tail suggests it may be a retention of a pleisiomorphic character state. It should be noted, however, that several proglacial lakes located near the Pittsburgh River of the Pliocene were the predecessors of the present-day Great Lakes [6]. Historically, the Old Allegheny River, a northwest flowing river, occupied what is now the region of Lake Erie. Perhaps an ancestral form of *M. punctulatus* initially occupied these waters.

## 5. Conclusions

A new species, *M. parallelus*, is described from Presque Isle, Lake Erie, Pennsylvania. *Micropterus parallelus* is distinguished from all other existing species in the genus. Based on data collected from specimens captured at Long Point, Canada, there appears to be at least two species of *Micropterus* in Lake Erie or a single polymorphic one. Obviously, more extensive research, including molecular data on *Micropterus* in the Great Lakes, is needed, because if more than one species exists, it will demand different management strategies for this popular sport fish.

## Figures and Tables

**Figure 1 biology-13-00660-f001:**
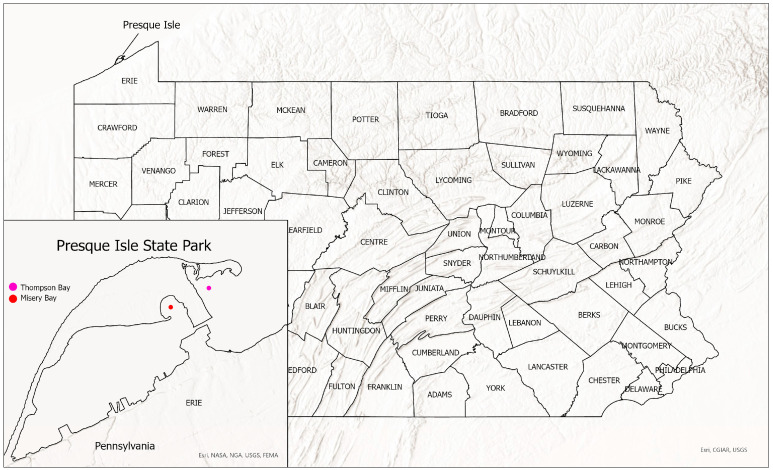
Presque Isle State Park showing location of Thompson Bay and Misery Bay.

**Figure 2 biology-13-00660-f002:**
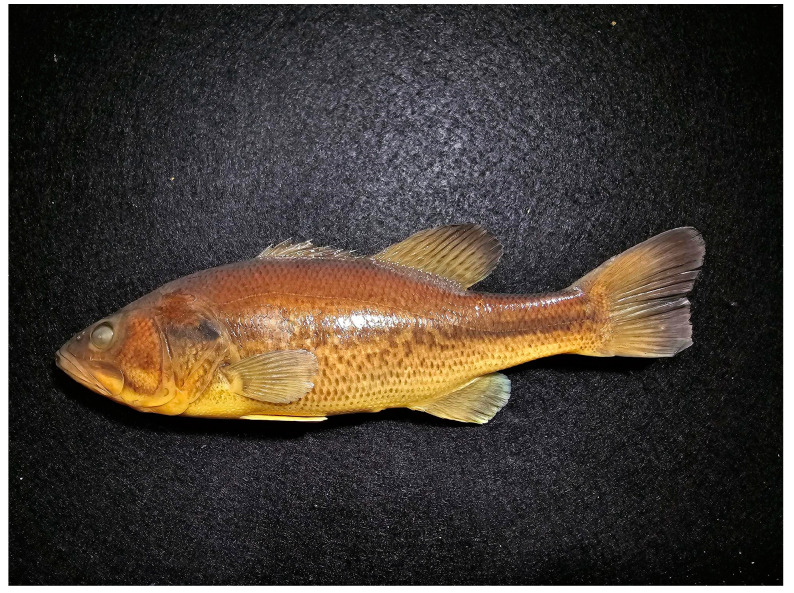
*Micropterus parallelus*. Holotype, PSU 7410, adult male 190.9 mm SL, Thompson Bay, Lake Erie, PA.

**Figure 3 biology-13-00660-f003:**
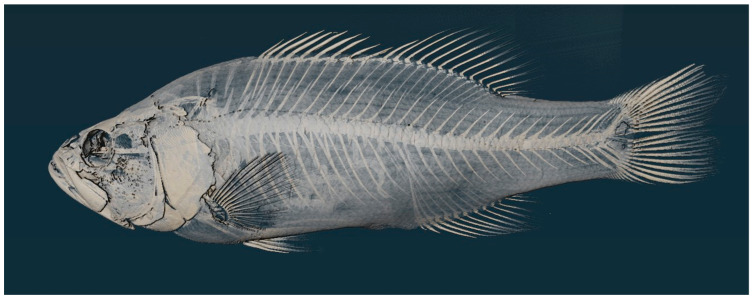
Micro CT scan of Holotype, PSU 7410, adult male 190.9 mm SL, Thompson Bay, Lake Erie, PA.

**Figure 4 biology-13-00660-f004:**
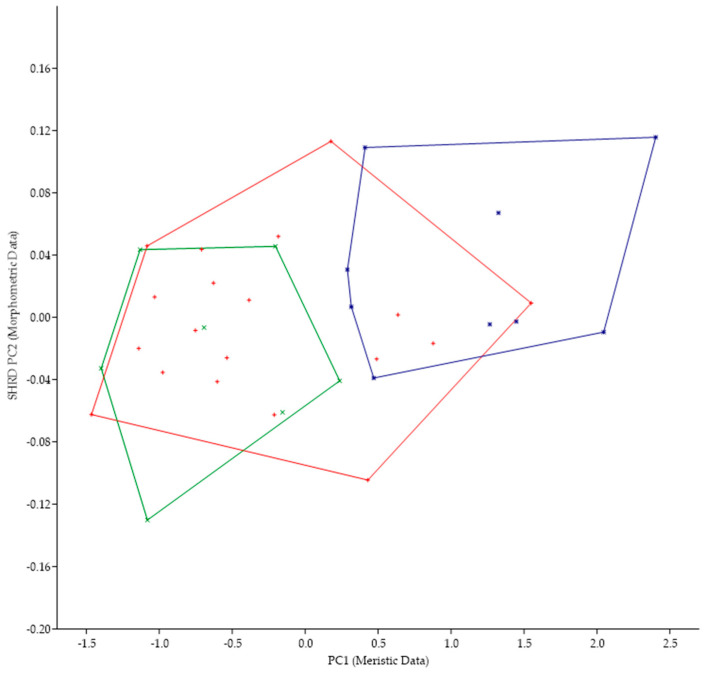
Plot of the second sheared principal components of morphometric data (SHRD PC2) and the first principal components of the meristic data (PC1) for *Micropterus parallelus* from Thompson Bay, PA (×), Misery Bay, PA (+), and *Micropterus* spp. Long Point, Canada (*).

**Figure 5 biology-13-00660-f005:**
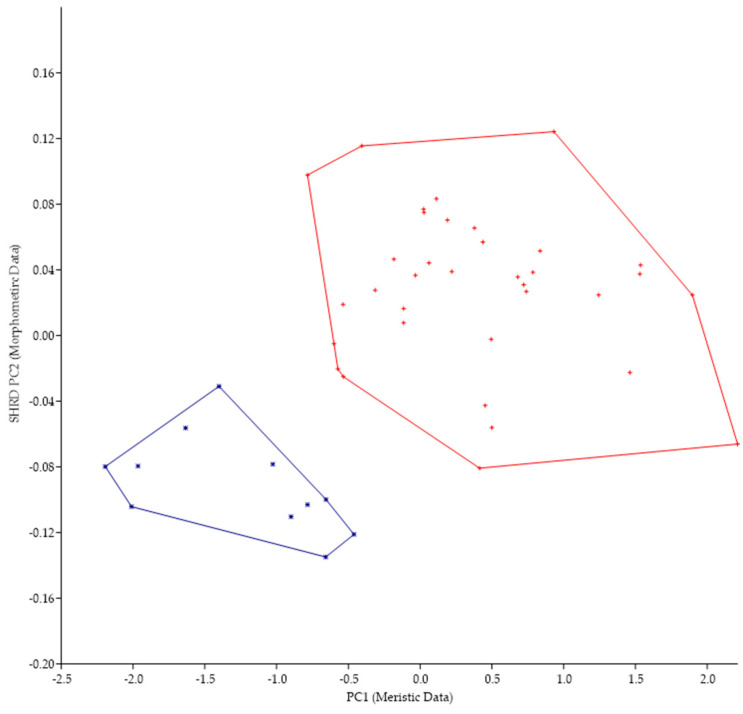
Plot of the second sheared principal components of morphometric data (SHRD PC2) and the first principal components of the meristic data (PC1) for *Micropterus parallelus* (+) and *Micropterus punctulatus* (*).

**Figure 6 biology-13-00660-f006:**
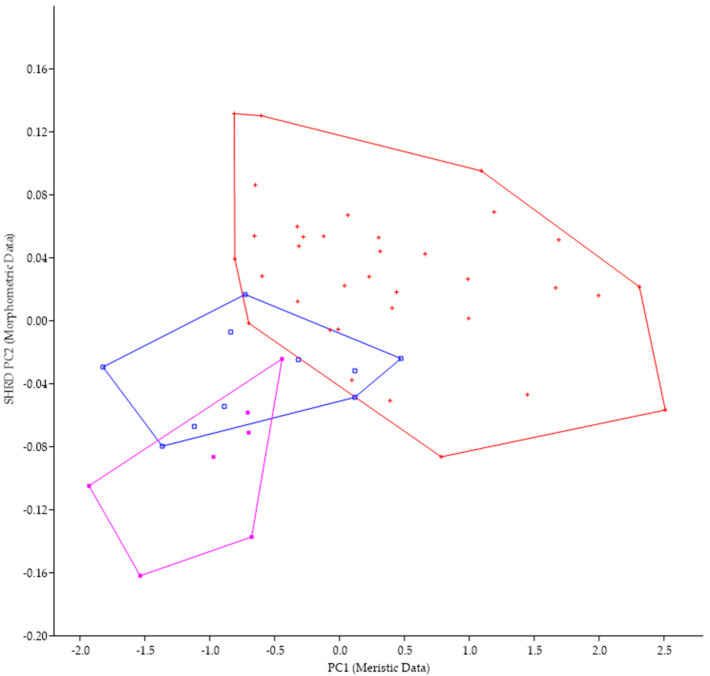
Plot of the second sheared principal components of morphometric data (SHRD PC2) and the first principal components of the meristic data (PC1) for *Micropterus parallelus* (+) and *Micropterus nigricans* from the Susquehanna River (□) and *M. nigricans* from the Ohio River Basin (■).

**Figure 7 biology-13-00660-f007:**
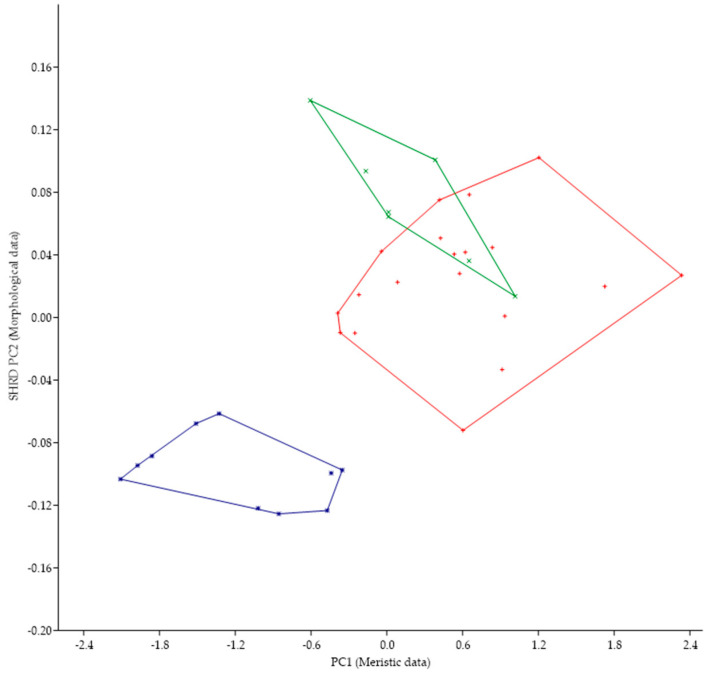
Plot of the second sheared principal components of morphometric data (SHRD PC2) and the first principal components of the meristic data (PC1) for Micropterus parallelus from Thompson Bay (×), from Misery Bay (+) and *Micropterus* spp. From Long Point, Canada (*).

**Table 1 biology-13-00660-t001:** Morphometric and meristic data for *Micropterus parallelus*. Data for fishes from Thompson Bay include the holotype.

Variable	Holotype	Thompson Bay (7)			Misery Bay (19)		
		Mean	SD	Range	Mean	SD	Range
Standard Length, mm	190.9	206.8	33.5	166.1–270.3	169.4	85.4	76.1–321.6
Head Length	66.7	72.3	12.1	64.2–94	59.7	31.4	27.7–111.0
**Percent Head Length**							
Snout Length	17.9	20.0	2.9	17.9–24.9	16.5	8.7	7.4–29.6
Post-Orbital Head Length	53.5	59.6	10.7	53.5–79	48.9	25.7	20.9–94.7
Horizontal Eye Diameter	11.4	11.4	1.8	10.9–14.5	10.1	3.9	6.1–17.3
Vertical Eye Diameter	9.8	9.9	1.9	9.4–13.4	9.1	3.5	5–15.2
Premaxillary Bone Length	29.9	32.5	6.3	29.7–43.8	29.2	16.1	11.9–57.6
Lower Jaw Length	35.2	35.1	6.3	32.3–46.8	31.1	17.2	13.3–61.6
Head Depth	33.6	30.0	5.7	25.4–35.4	23.0	11.4	10.4–40.3
Interorbital Width	18.0	18.8	3.0	15.9–23.7	15.8	8.4	6.7–28.8
**Percent Standard Length**							
Snout to Dorsal Fin Length	78.0	84.1	13.5	76.7–108.5	70.9	36.3	30.8–132.3
Snout to Pelvic Fin Length	67.2	71.2	10.8	64.3–91	58.5	30.2	24.3–111.3
1st Dorsal Fin Base Length	41.1	43.9	7.8	38.1–57.5	36.2	17.7	13.8–63.1
2nd Dorsal Fin Base Length	40.0	42.1	6.6	39.4–53.6	34.5	16.7	16.1–57.8
Ant. Dorsal to Ant. Anal	57.2	59.4	7.7	54.7–73.5	50.1	25.9	21.2–85.3
Ant. Dorsal to Post. Anal	61.9	64.5	9.6	60.1–82.3	54.0	26.8	23.4–89.9
Post. Dorsal to Ant. Anal	50.5	53.3	6.8	47.3–64.6	43.9	22.4	17.3–74.4
Post. Dorsal to Post. Anal	32.2	33.4	4.8	30–42	27.4	14.6	10.4–46.5
Post. Dorsal to Vent. Caudal	46.8	53.5	11.5	46.8–74.2	44.0	21.7	18.7–71.2
Post. Anal to Dorsal Caudal	46.0	57.0	13.5	46–72.1	42.1	21.1	18.4–71.4
Ant. Dorsal to Post. Peduncle	84.0	93.6	15.8	84–122.4	76.7	36.3	35–131.5
Post. Dorsal to Post. Peduncle	46.6	52.6	9.0	46.6–68.3	42.0	20.2	19.3–68.6
Caudal Peduncle Length	36.1	39.9	8.3	36.1–55.6	33.8	16.4	15–59.2
Least Caudal Peduncle Length	24.9	24.6	3.1	22.8–30.6	20.7	10.9	8.3–35.2
**Meristics**		**Mode**	**%Freq**	**Range**	**Mode**	**%Freq**	**Range**
Dorsal Spines	10.0	10	100		10.0	94.7	9–10
Dorsal Rays	13	13	85.7	12–13	11	57.8	11–13
Anal Spines	3	3	100		3	100	
Anal Rays	11	11	85.7	11–12	11	63.2	10–12
Lateral Line Scales	62	61/63	28.6	57–65	60–64	15.8	58–67
Cheek Row Scales	1	1	85.7	1–2	1	100	
Preoperculum Scale Rows	6	6	42.9	3–7	7	36.8	5–10
Operculum Scale Rows	7	4–6	26.7	4–7	6	47.4	5–8

**Table 2 biology-13-00660-t002:** Morphometric and meristic data for *Micropterus* spp. (*n* = 9) from Long Point, Canada (42.582877, −80.436170).

Variable			
	Mean	SD	Range
Standard Length, mm	260.4	37.5	208–307
Head Length	92.8	15.1	73.8–116.7
**Percent Head Length**			21–31.3
Snout Length	25.9	3.4	59.7–92.9
Post-Orbital Head Length	76.5	12.6	11.3–17
Horizontal Eye Diameter	13.8	1.8	10.4–15.8
Vertical Eye Diameter	12.9	1.8	35.5–58
Premaxillary Bone Length	44.6	7.4	37.5–60.4
Lower Jaw Length	48.2	8.3	29.3–47.4
Head Depth	38.5	6.2	20.2–31.2
Interorbital Width	25.5	3.9	21–31.3
**Percent Standard Length**			
Snout to Dorsal Fin Length	108.4	16.2	87–128.8
Snout to Pelvic Fin Length	90.7	13.2	72.9–109.1
1st Dorsal Fin Base Length	56.9	10.6	45–75.4
2nd Dorsal Fin Base Length	53.7	7.3	44–62.1
Ant. Dorsal to Ant. Anal	83.8	11.7	68.4–99.2
Ant. Dorsal to Post. Anal	84.8	11.2	68.6–98.4
Post. Dorsal to Ant. Anal	73.3	11.0	58.6–87
Post. Dorsal to Post. Anal	45.1	7.5	35.6–55.4
Post. Dorsal to Vent. Caudal	66.4	9.9	53.7–80.2
Post. Anal to Dorsal Caudal	66.7	9.2	52.7–75.7
Ant. Dorsal to Post. Peduncle	113.8	17.1	92.3–135.8
Post. Dorsal to Post. Peduncle	61.8	8.7	51.4–74
Caudal Peduncle Length	46.6	8.0	37.4–58.6
Least Caudal Peduncle Length	34.4	4.6	28–40.7
**Meristics**	**Mode**	**%Freq**	**Range**
Dorsal Spines	10	100	
Dorsal Rays	13	77.8	12–13
Anal Spines	3	100	
Anal Rays	12	55.6	11–12
Lateral Line Scales	63	33.3	59–65
Cheek Row Scales	2	100	
Preoperculum Scale Rows	8	33	6–10
Operculum Scale Rows	6/8	33.3	5–8

**Table 3 biology-13-00660-t003:** Morphometric and meristic data for *Micropterus punctulatus* (*n* = 10), *Micropterus nigricans* from the Allegheny (*n* = 10) and Susquehanna drainages (*n* = 7).

	*Micropterus punctulatus*			*Micropterus nigricans*(*Allegheny*)			*Micropterus nigricans (Sus.)*		
Variable	Mean	SD	Range	Mean	SD	Range	Mean	SD	Range
Standard Length, mm	94.7	26.8	70.9–157.3	120.7	31.7	78.4–157.8	162.3	42.6	121.8–264.1
Head Length	31.0	7.3	24.4–42.4	42.8	10.8	28.5–55.4	56.0	14.9	42.4–92
**Percent Head Length**									
Snout Length	29.5	1.1	27.4–30.7	27.5	2.0	24.9–31	27.7	1.5	26–30.8
Post-Orbital Head Length	82.4	5.4	75.6–91.4	78.5	3.4	73.2–84.3	85.3	4.3	76.9–90
Horizontal Eye Diameter	24.3	2.1	20.9–27.7	18.8	1.9	16.7–21.3	17.6	1.7	15.8–21.6
Vertical Eye Diameter	22.7	1.9	19.9–26.1	17.5	2.3	14.4–21	15.8	1.8	13.3–18.8
Premaxillary Bone Length	1.4	0.3	1.1–1.9	2.0	0.5	1.2–2.7	2.6	0.6	2.1–4
Lower Jaw Length	45.2	3.0	39.4–48.6	53.0	3.9	49–61.2	50.5	2.5	45.7–54.6
Head Depth	37.9	4.5	30.8–34.1	32.5	3.5	28.0–37.1	39.1	1.9	36.3–42.6
Interorbital Width	7.7	0.7	6.2–9.1	8.1	1.7	7.9–8.3	9.2	0.4	8.5–9.6
**Percent Standard Length**									
Snout to Dorsal Fin Length	41.7	3.1	33.2–44.1	41.6	1.2	40.6–44.2	42.1	1.0	40.8–43.8
Snout to Pelvic Fin Length	31.7	2.9	24.2–34.7	35.4	1.4	33.3–37.4	33.5	0.9	32.3–35.6
1st Dorsal Fin Base Length	20.8	1.9	16.3–22.3	20.8	1.3	18.8–22	22.7	1.8	19.5–25.2
2nd Dorsal Fin Base Length	17.8	1.5	14.4–19.4	20.2	0.6	19.4–20.9	20.5	1.1	19.2–22.1
Ant. Dorsal to Ant. Anal	24.9	1.6	21–27	28.4	1.2	27.2–30.7	28.5	1.2	26.5–30.5
Ant. Dorsal to Post. Anal	26.9	1.7	22.7–28.8	30.1	0.9	29.1–31.4	31.0	1.1	29.1–32.7
Post. Dorsal to Ant. Anal	23.1	1.8	19.4–25.6	25.3	0.9	24.3–26.8	25.1	0.7	24.1–26
Post. Dorsal to Post. Anal	13.8	1.1	11.5–15.3	14.9	0.5	14.3–15.4	15.2	0.7	13.6–16.2
Post. Dorsal to Vent. Caudal	24.1	2.1	18.4–25.6	23.3	0.8	22.6–24.9	23.2	1.1	21–25.2
Post. Anal to Dorsal Caudal	23.0	2.2	17.3–25.1	21.8	0.8	20.4–23	22.3	0.8	21.3–23.8
Ant. Dorsal to Post. Peduncle	40.6	3.0	33.1–43.7	42.2	1.1	40.9–43.8	42.2	1.0	40.7–44.1
Post. Dorsal to Post. Peduncle	23.4	2.3	17.8–26.4	22.9	1.5	20.7–25	22.6	1.5	20.4–25
Caudal Peduncle Length	18.0	2.1	12.6–20.1	17.6	1.2	16.2–19.4	18.1	1.2	16.5–20.1
Least Caudal Peduncle Length	11.1	0.8	9.5–12.3	11.9	0.4	11.5–12.5	12.3	0.5	11.5–13.2
**Meristics**	**Mode**	**%Freq**	**Range**	**Mode**	**%Freq**	**Range**	**Mode**	**%Freq**	**Range**
Anal Rays	10.0	90.0	10–11	10	70	10–11	11	85.7	10–11
Dorsal Rays	12.0	60.0	11–13	12	50	11–13	12	85.7	12–13
Dorsal Spines	10.0	100.0	NA	10	70	9–11	10	71.4	9–11
Anal Spines	3.0	100.0	NA	3	100		3	100	
Lateral Line Scales	61.0	30.0	60–68	60	70	59_63	61	42.8	59–62
Cheek Row Scales	1.0	100.0	NA	2	80	1–2	1	100	
Preoperculum Scale Rows	5–7	30.0	5–8	6	80	4–6	6	71.4	5–6
Operculum Scale Rows	6.0	40.0	3–6	6	50	5–8	4	57.1	4–5

## Data Availability

The CT data will be uploaded to Morphosource.

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
