# Peer review of "Morphological Analysis of a New Species of Micropterus (Teleostei: Centrarchidae) from Lake Erie, PA, USA"

_biology, 2024, doi:10.3390/biology13090660_

Round 1
Reviewer 1 Report (Previous Reviewer 1)
Comments and Suggestions for Authors
Dear Authors,
1. provide evidence/examples to readers that morphological analysis is really enough for new fish species description nowadays (references of at least two works 2020+), and not just several decades ago.
2. in material and methods part mention and/or discuss in discussion part possible formalin fixation impact on measurements of fish.
3. discuss in the discussion part about the necessity of genetic data for new fish species determination.
4. provide clarity to readers in material and methods part what fish meristics and morphometrics exactly were included in SPCA (synonym of SHRD?) and PCA, and why.
Comments based on particular manuscript parts:
Introduction
No main aim and tasks as separate paragraph in the end. Therefore, for readers it would be not clear enough how previous paragraphs connect with what, and why, you actually done.
Materials and Methods
No mention of SHRD.
Line 103: 27+9+10+10+7=62?
Results
Line 173 compare with Lines 103-108.
Line 191: based on what criteria?
Lines 255-257: where these values in Figure 4?
Line 258: what is SPCA2?
Table 3: Ohio -> Allegheny.
Discussion
Only general abstractions provided regarding species descriptions. Expand everything by adding information about the necessity of molecular data for new fish species determination. Provide clear picture for the readers what the currently accepted methods and techniques must be used in similar scientific research in 2024.
Lines 333-335: nothing changed from 2002?
Lines 344-345 that was in 1981. What is the current situation?
References
Most references are from 1980-1999 period. References after 2004 are rare, and only two references represent recent times (2016 and 2022, respectively). This do not provide clear picture for the readers what the currently accepted methods and techniques must be used in similar scientific research in 2024.
In reference list one reference without year information. Other errors, like bold text, exist. In other words, reference list is not prepared according journal requirements at the moment.
Author Response
- provide evidence/examples to readers that morphological analysis is really enough for new fish species description nowadays (references of at least two works 2020+), and not just several decades ago.
- I have cited two references from 2020 and 2023(
Dierickx, K, Snoeks, J. Protomelas krampus, a new paedophagous cichlid from Lake Malawi (Teleostei, Cichlidae). 2020. European J. Taxonomy. 672, 1–18.
Pauers, M.J., Phiri, T. B. Six new species of Labeotropheus (Cichliformes: Cichlidae) from the Malawian shore of Lake Malawi, Africa. 2023. Ichtyology and Herpetology, 2023, 264–29
- In material and methods part mention and/or discuss in discussion part possible formalin fixation impact on measurements of fish.
- This is a standard procedure for describing fishes from museum specimens. In any case, all the new species and comparative material were all formalin fixed museum specimens.
- discuss in the discussion part about the necessity of genetic data for new fish species determination.
- Genetic data are not necessary, and the two papers cited above descriptions of new species based solely on morphological data.
- provide clarity to readers in material and methods part what fish meristics and morphometrics exactly were included in SPCA (synonym of SHRD?) and PCA, and why.
- I describe in detail that a sheared PCA (SPCA) was used to analyze the morphometric data and a PCA was used to analyze the meristic data. At each point in the Results section, I describe which factors loaded most heavily on SPCA and PCA.
- No main aim and tasks as separate paragraph in the end. Therefore, for readers it would be not clear enough how previous paragraphs connect with what, and why, you actually done.
- I have moved the paragraph from the beginning of the introduction to the end that clarifies the purpose.
- I added more recent references and discussed why morphometric data is sufficient and essential for descriptions of species.
Reviewer 2 Report (Previous Reviewer 4)
Comments and Suggestions for Authors
The authors have addressed all my questions and concerns.
Author Response
No comments by reviewer
Reviewer 3 Report (Previous Reviewer 2)
Comments and Suggestions for Authors
There are some comments in the attached PDF file. Please revise this manuscript for the English language.

Author Response
I have checked the grammar and also reviewed the pdf. I have kept common names in capital letters, because they are proper nouns. Also, I kept the complete spelling of the generic name, the first time the scientific name of a species was used. Thereafter, I abbreviated the generic name, with the exception that I used complete spelling if it was the first word in a sentence.
This manuscript is a resubmission of an earlier submission. The following is a list of the peer review reports and author responses from that submission.
Round 1
Reviewer 1 Report
Comments and Suggestions for Authors
Dear Authors,
As I understand, you carried out quite hard work by measuring each fish various morphological parameters, and later by analyzing available data and presenting your results and findings in one manuscript. I tried to be objective and provide constructive comments and suggestions (see below) to you so the quality of your manuscript would be obviously better before publishing. In general, I am convinced that your research could be published in this scientific journal after major (actually, it could be major or minor revision, depending on your choices how to revise everything) revision. Most importantly, you should provide evidence/examples to readers that morphological analysis is really enough for new fish species description, discuss possible formalin fixation impact on measurements of fish, and expand discussion based on presented comments/suggestions.
Title
I think that this title could stay as it is. Alternatively, word ‘description’ could be removed or part ‘using morphological analysis’ added.
Simple Summary
In my opinion, it is too short for non-specialists to understand how everything was done during the study. New Micropterus species latin name should be mentioned here. Therefore, I advise to write 1-2 additional sentences in simple summary.
Abstract
Similar situation as with Simple Summary. I advise to write 1-3 additional sentences, and new Micropterus species latin name should be mentioned.
Introduction
To be honest, it was quite hard to read this part of the manuscript. The main reason was unnecessary confusion within and among paragraphs. In addition, it is not clear regarding introductions (where natural and where introduced populations?) and hybridization (natural and caused by anthropogenic activities). In some places there could be more references. Therefore, my advice to authors is to greatly modify or even rewrite this part of the manuscript. My suggestion how this important great modification of the manuscript could be done without necessity to rewrite everything, presented below, as well in the comments concerning particular Lines.
Write new Paragraph 1 with 1-2 sentences that mentions taxonomy and/or phylogenetic review problem. Then 1 additional sentence that this problem is also apparent in Micropterus genus. Current Paragraphs 2 (Lines 26-36) and 7 (Lines 77-82) should be combined to one new Paragraph, with small modifications, and then placed before current Paragraph 1 (Lines 19-25), i.e. this will be new Paragraph 2. Last current Paragraph 8 (Lines 83-89) should be rewritten or expanded and modified so readers could clearly understand introduction of Bass species/populations and hybridization problematics, especially in order to distinguish different species individuals without DNA investigations. Finally, this rewritten Paragraph 8 should end 1-2 sentences that describe object and aims of your study/article.
Paragraph 1 (Lines 19-25): Elaborate to readers why Micropterus genus is grouped together with other genus of Lepominae.
Paragraph 2 (Lines 26-36): Really confusing. Should be rewritten so readers could understand it easier. M. nigricans = M. salmoides = M. floridanus? Is that right or not?
Paragraph 3 (Lines 37-48): Lines 42-48 transfer to Line 24, before last sentence of Paragraph 1.
Paragraph 4 (Lines 49-55): Combine with Paragraph 3, after Lines 37-42 (after word ‘commonwealth’).
Paragraph 5 (Lines 56-66): In general, good.
Paragraph 6 (Lines 67-76): In general, good. In this paragraph revealed ethology of this fish species (spotted bass). It would be interesting to compare differences of ethology among largemouth bass and spotted bass but in the paragraphs about largemouth bass presented different information about species. I advise to write such information in the discussion together with part about M. parallelus ethology, and or future research in this area.
Paragraph 7 (Lines 77-82): Alabama bass latin name should be provided.
Paragraph 8 (Lines 83-89): Need to rewrite.
Line 24: Elaborate what do you mean by primitive.
Lines 26-27: Maybe it should be revised that it would be clear that they found that a nomenclatural revision was necessary but advised/suggested to delimitate Bass species to 19?
Lines 35-36: But called not M. nigricans but M. salmoides?
Lines 40-41: Indigenous besides introduction or only after introduction? Clarify.
Lines 83-84: Same species, yet different populations, or two different species? Clarify to readers. Also provide more information regarding introduction nuances.
Materials and Methods
In general, main questions are presented in the comments below. Also I advise provide more clarity to readers regarding your principal component analysis calculations, as now it is not clear what exactly was done and why.
Lines 91-92: Write what presupposed Bass species individuals were these 62 fish. M. haiaka or M. nigricans or M. punctulatus, or several different species? Clarify to readers, as it is your main research object and studied material.
Lines 92-93: Inform readers whether such procedures affect fish size, or other differences, compared to measurements of fresh fish or not.
Lines 102-103: Unnecessary, as it is a repetition of Lines 93-94. Modify or remove.
Lines 103-104: Any attempt to measure few same fish both sides and compare obtained results?
Results
In general, this part of the manuscript could be presented better. Try to correct typing errors. It would be not clear to readers how particular values appear in the text and how they were obtained. Even so, assuming that everything was calculated right, there is no doubt that M. parallelus is different species from M. nigricans.
Line 148: To readers it would be not clear what is this M. parallelus, as it is its first mention in the text. Write 1-2 sentences before Line 148 so readers can clearly understand everything regarding your main results.
Lines 151-152: Holotype of which species? It is M. parallelus, right? Then I advise to write information about holotype before Lines 149-150 to avoid any unnecessary confusion to readers.
Lines 153-155: Paratypes were of which species? M. salmoides and M. nigricans? Write clearly so readers can understand. Codes of individuals are necessary but without Supplementary Material means little to readers. Consider modify Lines 153-155 by writing additional information, and by dividing current sentence to at least two sentences.
Line 202: Number 9 change to n = 9.
Discussion
It is too short and important things are not discussed here. It is necessary that there would be at least two references of similar works, i.e. how particular fish species, especially cryptic, was defined using only morphological criteria/parameters. Also how measurements could or could not be affected by formalin fixation compared to fresh species measurements. If just morphological analysis for species determination is not enough then you must discuss it as well. Maybe it worth to create additional Table in discussion where 3-4 different Bass species (represented by 1 individual per species; including M. parallelus holotype) morphometrics would be presented and discussed? Part about morphological analysis with/without/vs genetic data for new fish species detection/description is clearly missing. Finally, M. parallelus ethology and possible hybridization, as well as guidelines for future research, should be discussed.
Lines 308-312: So, you suggesting two new species or just one? Expand discussion.
Conclusions
In general, it could be left as it is but then it would lack clarity, as proof was given that M. parallelus really differs morphologically from few studied Bass species but not all known Bass (Micropterus genus) species.
References
It is necessary to normalize everything both in the text (now in most cases is [] but in some cases scientists names, especially in Material and methods part) and in the reference list (sometimes bold and etc). I also advise you to add more new references of recent studies/reviews to support claims/findings of your study.
Author Response
- Changed title as suggested
- Did not use new name in abstract as I believe the name should not be used until after the description. If abstract is published separately then the new name becomes a nomen nudum.
- Added sentence to simple summary.
- I added that Lepomis also has 3 anal-fin spines.
- I rewrote the paragraph to better explain the nomenclatural changes.
- I provided the sci. name for Alabama Bass.
- I eliminated para 8 since it did not provide useful information.
- I replaced primitive with ancestral.
- I clarified where it has been introduced.
- I put M. parallelus in bold so it is clear that the holotype is of this species.
- I expanded the conclusions.
Reviewer 2 Report
Comments and Suggestions for Authors
Please revise the paper grammatically. The other important comments are in the attached PDF file.

Please revise the paper grammatically.
Author Response
- I changed keywords
- Common names are proper nouns so I kept them capitalized.
- I chose not to begin a sentence with an abbreviation
Reviewer 3 Report
Comments and Suggestions for Authors
The authors described a new species of Micropterus from Lake Erie, USA using morphometric and meristic data. The overall analysis of the data used is relevant. However, the major limitation is that they haven't used genetic data for the species delimitation. Many taxonomic studies on the genus Micropterus have employed both morphological and genetic data. DNA sequences are available for sister species of the new species described in this manuscript. Therefore, I strongly recommend the authors consider sequencing some gene sequences and run phylogenetic analysis and calculate pairwise genetic distances among allied species.
Following are some general comments.
The introduction is open-ended. It lacks the research question/s. Please consider revising it.
The materials and methods section has information about how many collections from which locality were analyzed. However, it lacks information about how those collections were made.
The methods section requires further clarification about how were the measurements made, what were the measurement accuracies,
Line 102-103: A total of 24 measurements and 15 counts (Konings and Stauffer 2006) were taken for 102 each fish. ................ repeated from lines 93-94.
Figure 1 requires a scale beside the specimen.
Author Response
- We took the specimens from the Penn State Univ. Fish Museum.
- eliminated the second use of A total of 24 measurements and 15 counts (Konings and Stauffer 2006) were taken for 102 each fish.
- scale is not needed since I provide the length of the fish.
Reviewer 4 Report
Comments and Suggestions for Authors
Ross et al. described a new species of Micropterus using morphometrics, and the morphological traits presented in the results appear promising. However, without molecular data and direct comparisons with sympatric Micropterus species, it is difficult to conclusively determine that this is indeed a new species. Additionally, a distribution map for each species should be included to aid readers unfamiliar with the genus. Further molecular data should also be provided to better delimit the species.
Please see the detailed comments below for more information.
------------------------
Specific Comments:
------------------------
Major comments:
1. Introduction: The introduction effectively outlines the taxonomy and general distribution of the genus Micropterus. However, it would be beneficial to include specific information on the genus in the northeastern and Midwest regions, especially considering the potential complexities introduced by artificial distributions. Previous studies have noted distinctions between species in these regions compared to southern species. This study could focus on comparison between the northeastern/Midwest species (or those in the Great Lakes region) and the newly proposed species.
Additionally, the current presentation of information is overwhelming and could be greatly enhanced by the inclusion of a map to visualize the distribution of species discussed in this study. A supplementary table detailing specimen information (such as location, collection time, and developmental stage) is needed. It would be particularly helpful to visualize the locations of specimens used in this study on the same map.
2. Methods: While extracting DNA from formalin-fixed samples is challenging, there are several kits available that facilitate obtaining high-quality DNA from such samples. Additionally, considering that the species involved are commonly targeted in recreational fishing, securing fresh or ethanol-preserved samples from natural history museums should be feasible. The discovery of a new species in a well-documented area is indeed exciting. However, to conclusively establish this as a distinct species, more comprehensive evidence is needed to eliminate other possibilities, such as polymorphism or hybridization.
Author Response
We provide a map of the new species. I explain why morphological analysis is sufficient to describe a new species and provide new references.
Round 2
Reviewer 1 Report
Comments and Suggestions for Authors
Dear Authors,
Your manuscript clearly require major revision and answers to almost all previously given constructive comments/suggestions to you. Most importantly, you should provide evidence/examples to readers that morphological analysis is really enough for new fish species description nowadays (references of at least two works 2020+), discuss possible formalin fixation impact on measurements of fish, and expand discussion based on presented comments/suggestions.
Introduction
In general, no major changes carried out here. It is still quite hard to read this part of the manuscript. The main reason: unnecessary confusion within and among paragraphs. In addition, it is not clear regarding hybridization (natural and caused by anthropogenic activities). You still need greatly modify or even rewrite this part of the manuscript. My suggestion how this could be done presented in previous review.
Materials and Methods
Main questions are presented in the comments below. Provide more clarity to readers regarding your principal component analysis calculations, as now it is not clear what exactly was done and why. Inform readers whether such procedures (formaline fixation) affect fish size, or other differences, compared to measurements of fresh fish or not. No information provided if there was any attempt to measure few same fish both sides and compare obtained results.
Results
Almost no changes in this part of the manuscript. This part could be presented better. It is not clear to readers how particular values appear in the text and how they were obtained.
Discussion
Not one change in this part while it was noted that this part of the manuscript cannot stay as it is without major revision. Important things are not discussed here. It is necessary that there would be at least two references of similar works, i.e. how particular fish species, especially cryptic, was defined using only morphological criteria/parameters. Also how measurements could or could not be affected by formalin fixation compared to fresh species measurements. If just morphological analysis for species determination is not enough then you must discuss it as well.
Part about morphological analysis with/without/vs genetic data for new fish species detection/description is clearly missing. Finally, M. parallelus ethology and possible hybridization, as well as guidelines for future research, should be discussed.
Conclusions
Not clear why this part was greatly expanded with citations (should there be any citations in conclusions?) when it was not required while discussion part was left as it is when it was necessary to rewrite/expand it.
Author Response
- reworked the introduction and gave examples of how morphological data are used to determine evolutionary relationships and clarified nomenclatural changes.
- expanded explanation of analyses used to distinguish between species.
- added explanation of species concepts and why we use morphological data to describe species.
Reviewer 3 Report
Comments and Suggestions for Authors
The major concern of the manuscript has not been addressed. I fear describing a new species in the present context without any support from the molecular data.
I suggested authors use a scale in the photograph of the holotype specimen, however, they replied that they mentioned the full length of the specimen in the text. Full length alone is not the single attribute that the readers want to know when they look at an image with a taxonomic description of a species.
The authors added two paragraphs in the conclusion section which do not read as concluding statements. It is just an addition to the text trying to justify why taxonomic identification is essential and historically species were described based on morphometric data, citing the literature of the 2000s.
I am sorry that I can not recommend it for publication at this stage in the journal Biology.
Author Response
- I cited works that use phenotypic data to describe species. The use of phenotypic data permit the designation of synapomorphies and autapomorphies that are lacking many times with the use of molecular data.
- I moved and expanded on the discussion so that it justifies the use of morphological data in the designation of species.
- A scale is not needed on the figure, since I include the SL in caption
Reviewer 4 Report
Comments and Suggestions for Authors
The authors have addressed most of my concerns. I only have a few minor comments:
Please visualize all the specimens (including other species) used in this study on the map.
The authors could add a paragraph stating future work on molecular phylogeny is needed to futhur confirm the species delimilation.
Author Response
1 I added the need for molecular data in the conclusion